# Ferrocene-Based Conjugated Microporous Polymers Derived from Yamamoto Coupling for Gas Storage and Dye Removal

**DOI:** 10.3390/polym12030719

**Published:** 2020-03-24

**Authors:** Zhiqiang Tan, Huimin Su, Yiwen Guo, Huan Liu, Bo Liao, Abid Muhammad Amin, Qingquan Liu

**Affiliations:** 1School of Materials Science and Engineering; Hunan Provincial Key Laboratory of Advanced Materials for New Energy Storage and Conversion; Hunan Provincial Key Laboratory of Controllable Preparation and Functional Application of Fine Polymers, Hunan University of Science and Technology, Xiangtan 411201, China; tanzq321@126.com (Z.T.); suhuiminss@126.com (H.S.); gyw98@126.com (Y.G.); hliu@hnust.edu.cn (H.L.); lb@hnust.edu.cn (B.L.); 2Department of Chemistry, University of Sahiwal, Sahiwal 38850, Pakistan; mabiduet@gmail.com

**Keywords:** conjugated microporous polymers (CMPs), ferrocene, gas adsorption, dye removal

## Abstract

Conjugated microporous polymers (CMPs) have conjugated skeleton and permanent porosity, and exhibit huge potential in developing novel functional materials for resolving the challenging energy and environment issues. Metal-containing CMPs often exhibited unique properties. In the present manuscript, ferrocene-based conjugated microporous polymers (FcCMPs) were designed and synthesized with 1,1′-dibromoferrocene and 5,10,15,20-Tetrakis(4- bromophenyl) porphyrin (FcCMP-1) or Tetra (*p*-bromophenyl) methane (FcCMP-2) as building units via Yamamoto coupling. FcCMPs were amorphous, and exhibited excellent thermal and physicochemical stability. The BET surface area of FcCMP-1 and FcCMP-2 was 638 m^2^/g and 422 m^2^/g, respectively. In comparison with FcCMP-2, FcCMP-1 displayed better gas storage capacity due to higher porosity. FcCMPs were also used as an adsorbent for removal of methyl violet from aqueous solution, and exhibited excellent adsorption properties due to the interaction between electron-rich conjugated structure of the polymers and methyl violet with cationic groups. Moreover, FcCMPs could be extracted and regenerated by an eluent and then re-used for high efficient removal of methyl violet.

## 1. Introduction

Conjugated microporous polymers (CMPs) are a kind of amorphous polymers with a three-dimensional network structure, which possess π-conjugated skeletons and permanent nanopores. The unique structure are unavailable in other porous polymers, which are usually not π-conjugated. CMPs have the characteristics of high specific surface area, light-weight, and attracted much attention in the past decade for great potential application in gas uptake [1,2], gas separations [3,4], and gas storage [5,6] as well as in photoredox catalysts [7,8,9]. From simple benzene ring, polycyclic aromatics, styryl derivatives to heterocyclic compounds can be employed as the building units, and active groups include bromine, iodine, boric acid, cyano, amino, aldehyde, and acetylene et al. Therefore, a variety of building units and extensiveness of active groups dramatically improved the design flexibility of CMPs skeleton and pore structure. This fact makes CMPs one of the most important platforms for developing novel organic porous polymers.

Synthetic routes of linear conjugated polymers can also be utilized for the preparation of CMPs, such as Suzuki coupling [10], Yamamoto coupling [11], Sonogashira coupling [2], and as well as trimerizations of acetylinic compounds [12]. The π-conjugated skeleton of CMPs can be further decorated by functionalization or metalation to improve its performance. For example, groups led by Weber [13] and Han [14] functionalized phenylethynyl CMPs with carboxyl groups. Zhu et al. [15] introduced various metal ions like lithium, sodium, potassium, and magnesium into carboxylic CMPs, and the loading amount of metal ion can reach as high as 2.6 mmol/g. Deng et al. [16] found that the hydrogen storage capacity of CMPs could be greatly enhanced by doping CMPs with Lithium ions. There are a few works reported that gas storage capacity could be improved by decorating CMPs with various metals [17,18,19,20]. 

Although meaningful progress has been made for the preparation, functionalization, and application of CMPs in the past decade, it is still an important scientific issue that based on the mind of block assembling, novel CMPs have to be further developed for extending CMPs functionalization, enriching CMPs kinds, and improving CMPs porosity [1]. Yamamoto coupling is an interesting synthetic route to generate CMPs with conjugated monomers with halogen groups as the building units. The coupling is usually performed with transition metal complex like Bis(1,5-cyclooctadiene)nickel(0) as a catalyst, and can generate carbon-carbon bonds between aryl-halogenide compounds by removing halogen substituents. Thus, the diversity of halogen monomers opened a door to design and prepare CMPs with various structures via Yamamoto coupling. 

Post-metalation of CMPs involves complicated processes and is generally achieved at the expense of porosity. Ferrocene and its derivatives are a kind of special organometallic compounds, and often used for designing polymers with special properties. With these in mind, ion-containing CMPs were designed and prepared by Yamamoto coupling of 1,1’-dibromoferrocene and 5,10,15,20-Tetrakis(4-bromophenyl)porphyrin or Tetra(p-bromophenyl)methane. The gas storage capacity and removal efficiency of dye from aqueous solution of ion-containing CMPs were investigated.

## 2. Experimental Section

### 2.1. Materials

Tetrahydrofuran (THF) and dimethylformamide (DMF) were purified by distillation with CaH_2_. Tetra(4-bromophenyl)methane (TBPM) was prepared with slight modification according to the method described in the literature [21], and 5,10,15,20-Tetra(4-bromophenyl)porphyrin (TBPP) was synthesized by one-step procedure from 4-brombenzaldehyde and pyridine according to the method from Alder-Longo [22]. The synthetic procedure and ^1^H NMR spectrum of TBPP and TBPM were shown in Appendix A. Ni(COD)_2_, 2,2’-bipyridyl, 1,5-cyclooctadiene, 1,1’-dibromoferrocene (FcBr), and various solvents were all purchased from commercial sources (Aladdin, Sigma-Aldrich, Shanghai, China) and used without further purification. 

### 2.2. Synthesis of FcCMPs

The synthesis procedure was slightly changed according to our previous work [2], and all reactions were performed under an argon atmosphere. A solution of TBPP (243.4 mg, 0.26 mmol) or TBPM (166.5 mg, 0.26 mmol), FcBr (180.0 mg,0.52 mmol), 1,5-cyclooctadiene (0.45 mL), Ni(COD)_2_ (1.0 g, 3.63 mmol), 2,2’-bipyridyl (0.57 g, 3.63 mmol), 36 mL dehydrated THF, and 24 mL DMF were added into the Schlenk flask, which was degassed at first, and equipped with a reflux condenser and a magnetic stir bar. The mixture was stirred at room temperature for 1 h, and then heated to 70 °C for reflux overnight. The suspension was cooled in an ice bath, and concentrated HCl was dropwise introduced into the suspension. The mixture was stirred for another 6 h. The precipitate was separated by filtration, and washed with H_2_O, CH_2_Cl_2_ and THF, respectively. Finally, the products were dried in vacuo to get ferrocene-based conjugated microporous polymers (FcCMPs) as purple powders.

### 2.3. Characterization

FT-IR spectra were collected in transmission mode in KBr pellets at room temperature on an Niclet 6700 spectrometer (Thermo Nicolet Corp., Waltham, MA, USA).

^13^C cross-polarization, magic angle spinning (CP/MAS) nuclear magnetic resonance spectra were recorded on an AVANCE III 400 spectrometer (Bruker NioSpin AG Facilities, Fällanden, Switzerland) equipped with a 4-mm HXY T3 PENCIL probe.

The thermogravimetric analysis (TGA) measurements were carried out under N_2_ flow on a Mettler TGA/SDTA 851 thermogravimetric analysis instrument from Mettler-Toledo Inc. of Switzerland. Samples were heated at a heating rate of 5 °C/min from 35 to 1000 °C in an N_2_ flow.

The scanning electron microscopy (SEM) analysis was performed on a model JSM-6700F scanning electron microscope (JEOL, Tokyo, Japan) and iridium (IXRF Systems) software with an accelerating voltage of 15 kV.

The X-ray photoelectron spectroscopy (XPS) experiments were performed on a K-Alpha 1063 spectrometer (Thermo Fisher Scientific, East Grinstead, UK) and the core level spectra were measured using a monochromatic Al Kα X-ray source (hv = 1386.6 eV).

X-ray diffraction curves of FcCMPs were determined on a D/max-rA XRD instrument (Rigaku Corp.,Tokyo, Japan) using a copper X-ray source.

Nitrogen sorption porosimetry was performed on a Micromeritics ASAP 2020 volumetric adsorption analyzer from Micromeritics Instrument Corp. of America. The experiments were carried out at the temperature of liquid nitrogen (77.3 K). The samples were first heated in a tube under vacuum at 120 °C for 20 h to remove adsorbed materials. Pore size distribution curves and the pore volumes of FcCMPs were derived from the adsorption branches of the isotherms using the non-local density functional theory (NLDFT).

Uptakes of H_2_, CO_2_ and CH_4_ of FcCMPs were also performed on the ASAP 2020 analyzer at different temperatures. 

A Lambd35 ultraviolet spectrophotometer (Perkin-Elmer Corp., Norwalk, Conn., US) was employed for determining the concentration of methyl violet.

NORWALK, CONN.

## 3. Results and Discussion

### 3.1. Chemical Structure of FcCMPs

The synthesis routes of FcCMPs were shown in Scheme 1. The chemical structure of FcCMPs was characterized by solid-state CP/MAS ^13^C NMR, Fourier transform infrared (FT-IR) spectroscopy and X-ray photoelectron spectroscopy (XPS), respectively. 

The CP/MAS ^13^C NMR spectra of FcCMP-1 and FcCMP-2 with assignment of the resonances were shown in Figure 1. The peaks from 66.2 to 69.5 ppm proved the existence of ferrocene units in FcCMPs, which was in agreement with the results of Okada et al. [23] The signal at 120.5 ppm could be assigned to the quaternary carbon of the benzene ring, which was linked to ferrocene units in FcCMP-1. However, the signal in FcCMP-2 moved to 125.2 ppm due to different chemical environments. Two peaks at 140.3 and 131.1 ppm could be ascribed to phenylene linkages and porphyrin skeleton of FcCMP-1. The signal at 151.3 ppm could be assigned to the carbons of C=N in porphyrin macrocycles. The peak at 50.8 ppm was attributed to the quaternary carbons in TBPM units of FcCMP-2. The peaks of 139.6 and 145.2 ppm were originated from quaternary and tertiary phenylene carbons in FcCMP-2. These results suggested that the spectra of FcCMPs were in accordance with their expected chemical structures as presented in Scheme 1.

FT-IR spectra (Figure 2a) revealed that the typical C-Br stretch vibration of FcCMPs at 875 cm^−1^ almost disappears in comparison with that of FcBr, indicating high conversion efficiency of FcBr. Absorption bands characteristic of bi-substituted ferrocene were observed at ~1000 cm^−1^, and the peaks at ~1400 cm^−1^ were attributed to C=C stretching vibration in cyclopentadiene rings, confirming that Yamamoto coupling has proceeded as expected. The absorption at ~2970 and ~3088 cm^−1^ was assigned to the stretching vibrations of unsaturated C–H bonds from phenyl and ferrocene units. X-ray photoelectron spectroscopy (XPS) was utilized to evaluate the elemental composition of FcCMPs, and the results were shown in the Figure 2b–d. Oxygen element was detected by XPS due to the fact that FcCMPs easily absorbs moisture from the air. In the XPS spectra, it could be observed trace bromine (0.12%–0.15%) derived from the unreacted end groups of FcBr. The nitrogen percentage (5.85%) in FcCMP-1 was essentially in agreement with the feed ratios of TBPP and FcBr. The binding energy at 708 and 720 eV corresponded to Fe 2P_3/2_ and Fe 2P_1/2_ bands in ferrocene units, respectively, which was in accordance with the results obtained by Gassman et al. [24]. However, the percentages of Fe element in both FcCMP-1 and FcCMP-2 were slightly lower than the theoretical values. TBPP and TBPM have higher coupling reaction efficiency than FcBr, which resulted in the main ends of the networks were terminated by TBPP and TBPM for Fc-CMP-1 and Fc-CMP-2, respectively. Overall, the analysis results of FT-IR and XPS spectra were consistent with that of ^13^C CP/MAS NMR, indicating that ferrocene building units were covalently integrated into the polymer networks.

### 3.2. Properties of FcCMPs

X-ray diffraction (XRD) patterns and TGA of FcCMPs were presented in the Figure 3. A dispersing and weak diffraction peak at 12.5° could be observed in XRD patterns of FcCMPs, suggesting an amorphous aggregation structure of the networks. Similar results have been reported by our group [20] and Yu et al. [17]. Due to its rigid structure, FcCMP-1 had excellent thermal stability in relative to most of CMPs, and began to lose its weight at a temperature below 400 °C. There was still 75.0 wt % residues as the temperature exceeded 800 °C. FcCMP-2 started to decompose above 300 °C, and its weight loss reached 32.0 wt % when the temperature achieved 800 °C. In comparison with FcCMP-2, FcCMP-1 had two-dimensional structure, and relatively stable chemical structure due to the conjugated porphyrin macrocycles, which could be responsible for the better thermal stability.

The morphology of FcCMPs were observed by SEM. As shown in Figure 4, FcCMP-1 was consisting of aggregated particles with diameters of 0.5–1 µm, and there are some interstitial voids that could produce some meso- and macropores. FcCMP-2 displayed aggregation of spherical particles with diameters of 1–2.5 µm. It was noted that the particle size was nonuniform, and there was also a large of interstitial voids between spherical particles.

### 3.3. Pore structure of FcCMPs

To evaluate the porosity parameters of FcCMPs, N_2_ adsorption and desorption isotherms were recorded at 77 K as shown in Figure 5. According to IUPAC classification [25], the isotherms of FcCMPs were type-I curves. The nitrogen uptake increased sharply at low relative pressure (*P*/*P*_0_ = 0–0.05), indicating the presence of substantial micropores in FcCMPs. The pore structure parameters were calculated and summarized in Table 1. The Brunauer–Emmett–Teller (BET) surface area of FcCMP-1 and FcCMP-2 are 638 and 422 m^2^/g, respectively. In relative to TBPM with a three-dimensional molecular structure, TBPP has a two-dimensional planar conjugated structure, which had a smaller steric hindrance and was favorable to couple with 1,1′-dibromoferrocene. The fact resulted in a higher crosslinking degree for FcCMP-1. Typically, the surface area of the product was increased along with the improved crosslinking degree, while the average pore size was declined. Obviously, higher surface area in the micropore region was achieved by FcCMP-1 in comparison with FcCMP-2. 

The pore size distribution curves were calculated from N_2_ adsorption isotherms by the non-local density-functional theory (NLDFT) model. The results exhibited FcCMPs had similar pore size distribution curves, which displayed a large population of pores in the micropore range extending to the lower part of the mesopore region. The surface area from the micropore region reached about 75% in the BET surface area of FcCMPs, which was in agreement with the results from the isotherms type and pore size distribution curves.

### 3.4. Gas Uptakes of FcCMPs 

Microporous properties and excellent chemical stability promoted us to apply FcCMPs for gas storage. H_2_ storage property of FcCMPs was performed under 77 K, and CO_2_ and CH_4_ adsorption/desorption isotherms were collected at 273 and 298 K, respectively. The gas uptakes were calculated and provided in Table 2. FcCMP-1 with a higher BET surface area exhibited a better gas capture property than FcCMP-1. The maximum CO_2_ and CH_4_ uptake of FcCMP-1 at 273 K/1 bar were 2.2 and 0.69 mmol/g, respectively, and the value of H_2_ was 5.5 mmol/g at 77 K/1 bar. The adsorption capacity and selectivity of CO_2_ over CH_4_ of FcCMP-1 is higher than that of FcCMP-2, which may be related to FcCMP-1 structures with incorporated Lewis basic N atoms enhancing the interaction between CO_2_ and polymer skeleton. The initial selectivity (CO_2_/N_2_) was determined to be 51.5 and 2.8 according to the simulation calculation, respectively.

Gas capture performance of FcCMPs and other porous organic polymers such as COFs [26,27,28], CMPs [18,22,29,30], and POPs [11,17,20,31,32,33] were summarized in Appendix A. In comparison with COFs, FcCMPs exhibited slightly higher CO_2_ capture capacity, and similar H_2_ and CH_4_ storage performance under the same conditions, although the BET surface area of COFs were substantially higher than those of FcCMPs. Obvious differences in the average pore diameter and the pore size distribution of FcCMPs and COFs were responsible for that. CMP showed similar porosity and gas capture performance with FcCMPs, but it should be noted that CMP loaded with lithium displayed H_2_ storage capacity as high as 30.5 mmol/g [18]. In relative to FcCMPs, most of POPs often had a higher crosslinking degree, greater BET surface area, and better gas uptake performance. However, FcCMP-1 displayed higher CO_2_ adsorption capacity than that of FPOP-2, although Fc-CMP-2 possessed a lower surface area and a similar structure with FPOP-2 [31]. 

In order to further study the gas adsorption performance of FcCMPs, the coverage-dependent isosteric heat of adsorption for CO_2_ and CH_4_ were simulated using the following expression:(1)$$\ln(P)=\ln(N)+\frac{1}{T}\sum_{i=0}^{m}a_{i}N^{i}+\sum_{j=0}^{n}b_{j}N^{j}$$
(2)$$Q_{st}=-R\sum_{i=0}^{m}a_{i}N^{i}$$

The physical quantities *P*, *N*, and *T* represent the pressure (torr), adsorption amount (mmol/g), and temperature (K), respectively. *a_i_* and *b_j_* are the virial coefficients, and the numbers are described by *m* and *n* (*m* and *n* no longer increase until the additional contribution of *a* and *b* are considered to be statistically insignificant for the overall fit, and the average value of the squared deviations from the experimental values is minimized). The gas adsorption data of FcCMPs collected at 273 K and 298 K were well fitted by Virial equation (*R^2^* > 0.99999), and the Virial coefficients of *a*_0_ to *a*_m_ could be obtained from the fitting curves. The values of *a*_0_ to *a*_m_ were substituted into the Equation (2) to calculate isosteric heats (*Q_st_*) of FcCMPs, and the results were provided in Figure 6 and Table 2. As shown in Figure 6, *Q_st_* values of CO_2_ and CH_4_ decreased with the increasing loading amount, indicating the interaction between gas molecules and polymer networks declining. Moreover, *Q_st_* values of FcCMPs were similar to those of other porous polymers such as hyper-crosslinked polymers (HCPs) [34,35], CMPs [5], microporous polyimides (MPIs) [36], and porous polymer networks (PPNs) [37].

### 3.5. Methyl Violet Adsorption of FcCMPs

In view of the excellent porosity of FcCMPs, methyl violet (MV) was selected as the target molecule to investigate dye adsorption capacity and cyclic adsorption efficiency of FcCMPs. FcCMPs (2.5 mg) were used for treating 10 mL of MV solution (67.0 mg/L). After the mixtures were ultrasonically dispersed for 5 min, the solution turned into colorless or light color, which indicated MV was successfully adsorbed by FcCMPs. The filtrates were collected by sintered funnel. MV concentration of the collected filtrates were analyzed and calculated by UV spectra according to a standard curve. Moreover, FcCMPs were extracted and regenerated by the eluent of MW14 reported in our previous works [38,39], and were dried for the next adsorption cycle. The above process was repeated three times to investigate the reusability of FcCMPs. UV spectra of the original and collected solutions were provided in Figure 7.

The results indicated that after the first cycle, about 98.2% and 97.7% MV in aqueous solution could be removed by FcCMP-1 and FcCMP-2, respectively. We think the high removal efficiency of FcCMPs should be attributed to numerous accessible nanoporous channels and electron-rich conjugated structure of the networks, which could generate an interaction with the cationic dye of MV. The residual MV content of the filtrates increased slightly with increasing adsorption cycles, however, after the third cycle, more than 96% MV could still be effectively removed by FcCMPs. This fact confirmed that FcCMPs have excellent removal capacity of MV, and with the help of eluent MW14, FcCMPs exhibited satisfactory reusability for removing MV. 

## 4. Conclusions

Two nanoporous organic polymers of FcCMP-1 and FcCMP-2 were successfully synthesized by Yamamoto coupling of 1,1’-dibromoferrocene and polybrominated aromatic compounds with Ni(COD)_2_ as a catalyst. FcCMPs possessed excellent chemical and physical stability. FcCMPs exhibited moderate porosity with a BET surface area of 638 m^2^/g for FcCMP-1 and 422 m^2^/g for FcCMP-2, and micropores occupied most porosity of FcCMPs. In comparison with FcCMP-2, FcCMP-1 exhibited better gas storage capacity due to a higher BET surface area and pore volume. Application of FcCMPs was further extended for removal of methyl violet from aqueous solution, and the results suggested that FcCMPs had excellent MV adsorption performance due to the interaction between electron-rich conjugated structure of polymers and MV with the cationic group. Moreover, FcCMPs could be easily regenerated by an eluent and then re-used for high efficient removal of MV.

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
