# Peer review of "Ferrocene-Based Conjugated Microporous Polymers Derived from Yamamoto Coupling for Gas Storage and Dye Removal"

_polymers, 2020, doi:10.3390/polym12030719_

Round 1

Reviewer 1 Report

Author did very good study on the Ferrocene-based Conjugated Microporous Polymers for Gas Storage and Dye Removal. Though the gas adsorption is not outstanding, CMP containing ferrocene and porphyrin moieties are always interesting from many perspectives. I wish author should add a table to show the gas adsorption behaviour of the these CMPs with state of art POPs or COFs. A detailed discussion will be highly appreciated.

Author Response

  1. Author did very good study on theFerrocene-based Conjugated Microporous Polymers for Gas Storage and Dye Removal. Though the gas adsorption is not outstanding, CMP containing ferrocene and porphyrin moieties are always interesting from many perspectives. I wish author should add a table to show the gas adsorption behaviour of the these CMPs with state of art POPs or COFs. A detailed discussion will be highly appreciated.

Response 1: Due to too many POPs or COFs reported up to now, only porous polymers with ferrocene building units were listed to compare the gas storage behavior of FcCMPs, and a discussion was made.

Reviewer 2 Report

The manuscript deals with comprehensive characterization of 2 porous conjugated polymers based on ferrocene bacbone. It represents a continuation of previous work conducted by the same group (ref 2 & 11). Both obtained materials were porous as confirmed by good sorption of gases  such as N2, H2, CO2 and CH4.  Moreover, high sorption capacity towards methyl violet was demonstrated. Overall, the work seems to be competently done and thus it deserves publication in Polymers. However, some points should be addressed:

  1. The synthesis of building blocks TBPP and TBPM (described in Section 2.2) was accomplished based on the literature procedures and therefore there is no need to report it in the manuscript. Perhaps it could be provided as the Supplementary Material. The presentation of 1H NMR spectra of TBPP and TBPM is also not necessary.
  2. Lines 176-177: Based on analysis of TGA plots in Fig. 5, I would suggest to rephrase „FcCMP-2 started to decompose below 400 C” as „FcCMP-2 started to decompose above 300 C”.
  3. Line 201: Did the author mean NLDFT model?
  4. I would suggest to provide gas uptakes not only in wt% but also in mmol/g for easier comparison of sorption selectivities.
  5. The manuscript shows many language deficiencies. It should carefully checked and revised prior to final acceptance.

Author Response

 The manuscript deals with comprehensive characterization of 2 porous conjugated polymers based on ferrocene bacbone. It represents a continuation of previous work conducted by the same group (ref 2 & 11). Both obtained materials were porous as confirmed by good sorption of gases  such as N2, H2, CO2 and CH4.  Moreover, high sorption capacity towards methyl violet was demonstrated. Overall, the work seems to be competently done and thus it deserves publication in Polymers. However, some points should be addressed:

Point 1: The synthesis of building blocks TBPP and TBPM (described in Section 2.2) was accomplished based on the literature procedures and therefore there is no need to report it in the manuscript. Perhaps it could be provided as the Supplementary Material. The presentation of 1H NMR spectra of TBPP and TBPM is also not necessary.

Response 1: The synthesis procedure and the characterization data of building blocks TBPP and TBPM were moved to Supplementary Material.

Point 2:Lines 176-177: Based on analysis of TGA plots in Fig. 5, I would suggest to rephrase „FcCMP-2 started to decompose below 400 C” as „FcCMP-2 started to decompose above 300 C”.

Response 2: The sentences were rephrased as “FcCMP-1 had  began to lose its weight at a temperature below 400 oC; and FcCMP-2 started to decompose above 300 oC”.

Point 3: Line 201: Did the author mean NLDFT model?

Response 3: Sure, density-functional theory (DFT) model was replaced by “the non-local density- functional theory (NLDFT) model”

Point 4: I would suggest to provide gas uptakes not only in wt% but also in mmol/g for easier comparison of sorption selectivities.

Response 4: Gas uptakes of FcCMPs were provided in wt% and in mmol/g.

Point 5: The manuscript shows many language deficiencies. It should carefully checked and revised prior to final acceptance.

Response 5: The language deficiencies were carefully checked and revised.

Reviewer 3 Report

Review of ,,Ferrocene-based Conjugated Microporous Polymers Derived fromYamamoto Coupling for Gas Storage and Dye Removal’’.

Two nanoporous organic polymers of were synthesized by Yamamoto coupling of 1,1’-dibromoferrocene and polybrominated aromatic compounds with a catalyst. The authors provide a detailed description of its structure. But the article does not contain information about their use. Other doubts and questions are included below.

  1. Yamamoto Coupling - I think that this method of coupling should be discussed at the beginning and its mechanism.
  2. Please develop the concept ‘’Conjugated microporous polymers’’
  3. Line 32: ‘’application in gas adsorption [1,2]’’: which is exactly where? Please give more details.
  4. ,,..materials for resolving the challenging energy and enviroment issues’’. How do these materials solve the energy and environmental problem?
  5. The introduction does not show the topic sufficiently. There is a lack of information on porous materials. There is no comparison of the properties of products obtained with the Conjugated micropore polymers (CMPs) technique compared with other techniques. The CMPs technique should be further discussed. Discuss how long this technique is known and how it possibly changed. Why was CMPs chosen?
  6. The experimental part contains too few references.
  7. 4. Characterization- the addresses of the companies from which the devices used are missing.
  8. Line 185-186: How does the size and shape of the cells affect the properties of the resulting material.
  9. Where will this material (Microporous Polymers) be used?
  10. Line 169. Figure 4 : Four pictures in figure 4 should be marked 4a, 4b, 4c, 4d and discussed each separately for greater clarity.
  11. Line 192: Figure.7. Should be Fig. 7.
  12. Line 208, I didn’t notice in text description of abbreviations from Table 1.

Author Response

 Two nanoporous organic polymers of were synthesized by Yamamoto coupling of 1,1’-dibromoferrocene and polybrominated aromatic compounds with a catalyst. The authors provide a detailed description of its structure. But the article does not contain information about their use. Other doubts and questions are included below.

Point 1: Yamamoto Coupling - I think that this method of coupling should be discussed at the beginning and its mechanism.

Response 1: Yamamoto coupling was introduced in the third paragraph of Introduction.

Point 2: Please develop the concept ‘’Conjugated microporous polymers’’

Response 2: The concept of conjugated microporous polymers was developed in the first paragraph of Introduction.

Point 3: Line 32: ‘’application in gas adsorption [1,2]’’: which is exactly where? Please give more details.

Response 3: “application in gas adsorption” was replaced by “application in gas uptake, gas separation, and gas storage”.

Point 4: ,,..materials for resolving the challenging energy and enviroment issues’’. How do these materials solve the energy and environmental problem?

Response 4: CMPs are unique in that they are nanoporous and p-conjugated, whereas their structures can be designed at the molecular level and synthetically controlled. By virtue of high surface areas and microporous characteristics, CMPs have emerged as a new class of porous materials for gas adsorption and storage applications. The pores provide open space and are accessible to various guest molecules and metal ions, allowing for the construction of supramolecular structures and organic–inorganic hybrids. Most importantly, CMPs allow the complementary utilization of p-conjugated skeletons and nanopores for functional exploration. They have shown great potential for challenging energy and environmental issues, as exemplified by their excellent performance in such applications as gas adsorption,

heterogeneous catalysis, light emitting, light harvesting and electric energy storage.

Point 5: The introduction does not show the topic sufficiently. There is a lack of information on porous materials. There is no comparison of the properties of products obtained with the Conjugated micropore polymers (CMPs) technique compared with other techniques. The CMPs technique should be further discussed. Discuss how long this technique is known and how it possibly changed. Why was CMPs chosen?

Response 5: Introduction was re-organized and revised.

Point 6: The experimental part contains too few references.

Response 6: A reference was added in the synthetic procedure of CMPs.

Point 74. Characterization- the addresses of the companies from which the devices used are missing.

Response 7: The companies of the devices in Characterization were provided.

Point 8: Line 185-186: How does the size and shape of the cells affect the properties of the resulting material.

Response 8: In relative to TBPM with three-dimension molecular structure, TBPP has a two- dimensional planar conjugated structure, which had a smaller steric hindrance and was favorable to couple with with 1,1'-dibromoferrocene. The fact resulted in a higher crosslinking degree for FcCMP-1. Typically, the surface area of the product was increased along with the improved crosslinking degree, while the average pore size was declined.

Point 9: Where will this material (Microporous Polymers) be used?

Response 9: The microporous polymers were employed for gas storage and dye removal in the present manuscript.

Point 10: Line 169. Figure 4 : Four pictures in figure 4 should be marked 4a, 4b, 4c, 4d and discussed each separately for greater clarity.

Response 10: Four pictures has been marked as 4a, 4b, 4c, 4d in Figure 4 and in the manuscript.

Point 11: Line 192: Figure.7. Shoul d be Fig. 7.

Response 11: Figure.7. was changed to be Fig.7.

Point 12: Line 208, I didn’t notice in text description of abbreviations from Table 1.

Response 12: The abbreviation descriptions in Table 1 was provided.
